

# Quantifying the emission changes and associated air quality impacts during the COVID-19 pandemic in North China Plain: a response modeling study

Jia Xing[1,2], Siwei Li[3, 4, *], Yueqi Jiang[1,2], Shuxiao Wang[1,2,*], Dian Ding[1,2], Zhaoxin Dong[1,2], Yun Zhu[5], Jiming Hao[1,2]

[1] State Key Joint Laboratory of Environmental Simulation and Pollution Control, School of Environment, Tsinghua University, Beijing 100084, China

[2] State Environmental Protection Key Laboratory of Sources and Control of Air Pollution Complex, Beijing 100084, China

[3] School of Remote Sensing and Information Engineering, Wuhan University, Wuhan 430079, China

[4] State Key Laboratory of Information Engineering in Surveying, Mapping and Remote Sensing, Wuhan University, Wuhan 430079, China

[5] College of Environment and Energy, South China University of Technology, Guangzhou Higher Education Mega Center, Guangzhou 510006, China

*Corresponding Authors: Shuxiao Wang (email: shxwang@tsinghua.edu.cn; phone: +86-10-62771466; fax: +86-10-62773650); Siwei Li (email: siwei.li@whu.edu.cn)

## Abstract

Quantification of emission changes is a prerequisite for the assessment of control effectiveness in improving air quality. However, the traditional bottom-up method for characterizing emissions requires detailed investigation of emissions data (e.g., activity and other emission parameters) that usually takes months to perform and limits timely assessments. Here we propose a novel method to address this issue by using a response model that provides real-time estimation of emission changes based on air quality observations in combination with emission-concentration response functions derived from chemical transport modeling. We applied the new method to quantify the emission changes in the North China Plain (NCP) due to the COVID-19 pandemic shutdown, which overlapped the Spring Festival holiday. Results suggest that the anthropogenic emissions of $NO_2$, $SO_2$, VOC, and primary $PM_{2.5}$ in NCP were reduced by 51%, 28%, 67% and 63%, respectively, due to the COVID-19 shutdown, indicating longer and stronger shutdown effects in 2020 compared to the previous Spring Festival holiday. The reductions of VOC and





primary PM$_{2.5}$ emissions are generally effective in reducing O$_3$ and PM$_{2.5}$ concentrations. However, such
air quality improvements are largely offset by reductions in NO$_x$ emissions. NO$_x$ emission reductions lead
to increases in O$_3$ and PM$_{2.5}$ concentrations in NCP due to the strongly VOC-limited conditions in winter.
A strong NH$_3$-rich condition is also suggested from the air quality response to the substantial NO$_x$ emission
reduction. Well-designed control strategies are recommended based on the air quality response associated
with the unexpected emission changes during the COVID-19 period. In addition, our results demonstrate
that the new response-based inversion model can well capture emission changes based on variations in
ambient concentrations, and thereby illustrate the great potential for improving the accuracy and efficiency
of bottom-up emission inventory methods.

**Keywords:** emission changes, response model, ozone, PM$_{2.5}$, control effectiveness



## 1. Introduction

Accurate estimation of anthropogenic emissions is crucial for atmospheric modeling studies and provides the basis for developing effective air pollution controls (Wang et al., 2010). A comprehensive emission inventory consists of the emission rates of primary particulate matter components and gaseous pollutants and precursors that are allocated over time and space. These inventories are usually developed using bottom-up methods that gather detailed information about source activity and other emission parameters (Wang et al., 2011; Xing et al., 2015; Li et al., 2017). The challenge is that such investigation is costly and time consuming, and therefore the latest emission inventories usually lag current conditions by a year or more. Many studies also apply a top-down methods to constrain emission estimates using satellite retrievals and modeling methods (Tang et al., 2013, 2019; Lu et al., 2015; Miyazaki et al, 2017; Cao et al., 2018; Zhang et al., 2018). The top-down inversion method can well reflect the change in emissions in a timely manner, and thus efficiently estimate emissions at high spatial and temporal resolution to complement bottom-up inventories. Previous inversion studies have focused on individual pollutants that can be measured directly; however, studies are lacking that use top-down methods to estimate emissions of multiple pollutants, including those that cannot be directly measured, such as primary fine particular matter (p-PM$_{2.5}$).

The ongoing Coronavirus disease 2019 (COVID-19) pandemic has led to 4,600 deaths in mainland China (by May 24, 2020, https://news.google.com/covid19/), and has resulted in a dramatic curtailment of routine economic and social activities. The shutdown of human activities during the COVID-19 pandemic has led to reduced pollutant emissions and possibly improved air quality (Shi et al., 2020; Wang et al., 2020a). Yet according to ambient concentration measurements, heavy PM$_{2.5}$ pollution still occurred during the COVID-19 period, and formation of secondary pollutants was actually enhanced in China (Li et al., 2020; Huang et al., 2020). Some studies attributed pollution enhancements to atypical weather conditions that are favorable for air pollution formation (Wang et al., 2020b). Meanwhile, the unexpected





reduction of anthropogenic emissions due to the COVID-19 shutdown might vary significantly for
different sectors and species. For example, emissions from domestic sources might have increased due to
a greater demand for home heating and other essential consumptions during periods with stay-at-home
orders in effect. Moreover, the coincidence of the COVID-19 shutdown and the Spring Festival in China
resulted in large numbers of people confined to their rural or small-city hometowns, where consumption
patterns differ greatly from their primary residence in megacities. Relative to previous years, both
emissions and meteorological conditions varied simultaneously during the 2020 COVID-19 shutdown,
and an accurate estimation of the changes in anthropogenic emissions accounting for meteorological
variations is needed to characterize the impacts of COVID-19 on air quality.

Here we propose a novel inversion technique based on a multi-pollutant nonlinear response model

to estimate the emission changes in NCP during the COVID-19 shutdown. Emission changes for the
COVID-19 period are calculated as the difference between emission estimates for actual conditions and
hypothetical conditions assuming the shutdown did not occur. The hypothetical emissions are determined
by combining top-down emission estimates from before and after the shutdown with estimates of the
temporal variation in emissions from a bottom-up emission inventory. Additionally, we estimate the
change in emissions associated with the Spring Festival holiday in 2019 to contrast with results for the
combined Spring Festival holiday and COVID-19 shutdown in 2020. Finally, we evaluate the impacts on
$PM_{2.5}$ and $O_3$ concentrations of the combined emission changes and for each emitted species to provide
insights for the design of effective control strategies in the future.

## 2. Methods

**2.1 Response model to estimate the actual emissions from observed surface concentrations**

The principle of the new response-based inversion model (hereafter "the response model") is to

adjust the assumed prior emissions such that concentration predictions match observations.  The core
element of the inversion method is a nonlinear response surface model (RSM) that represents the emission-



concentration response functions. The framework of the response model is illustrated in Figure 1. We
conduct chemical transport model simulations using prior emissions to get the original simulated
concentrations of six pollutants (i.e., $NO_2$; $O_3$; $SO_2$; $PM_{2.5}$; sulfate, $SO_4^{2-}$; and nitrate, $NO_3^-$), as well as the
response functions derived from the RSM (Xing et al., 2011; Wang et al., 2011; Xing et al., 2017; 2018).
We then adjust the emission ratio of five pollutants (i.e., $NO_2$, VOC, $SO_2$, $NH_3$ and primary $PM_{2.5}$) to
estimate the updated simulated concentrations to match with the observations.

Based on our previous knowledge of emission-concentration response relationships, we first adjust

$NO_x$ emissions such that RSM predictions match $NO_2$ observations (see E1), since $NO_2$ concentrations
have a strong linear relationship with $NO_x$ emissions (Xing et al., 2017).
$$E'_{NOx} = r_{NOx} \times E^*_{NOx} = E^*_{NOx} \times \frac{C^o_{NO2}}{C^s_{NO2}} \qquad (E1)$$
where $E'_{NOx}$ is the adjusted $NO_x$ emissions; $E^*_{NOx}$ is the prior $NO_x$ emissions; $r_{NOx}$ is the adjustment ratio
for $NO_x$ emissions; $C^o_{NO2}$ is the observed $NO_2$ concentrations; and $C^s_{NO2}$ is the simulated $NO_2$
concentrations.

Next, we adjust VOC emissions such that RSM predictions match observed $O_3$ concentrations, since

$O_3$ concentrations are solely determined by VOC emissions after $NO_x$ emissions are determined in the
previous step. The adjusted VOC emission ratio (i.e., $r_{VOC} = E'_{VOC}/E^*_{VOC}$) is determined by solving the
following equation E2:
$$\Delta O_3 = (C^o_{O3} - C^s_{O3}) = RSM_{O3}(r_{NOx}, r_{VOC}) \qquad (E2)$$
where $E'_{VOC}$ is the adjusted VOC emissions; $E^*_{VOC}$ is the prior VOC emissions; $\Delta O_3$ is the difference
between observed $O_3$ concentrations ($C^o_{O3}$) and simulated $O_3$ concentrations ($C^s_{O3}$); and $RSM_{O3}$ is the
response function of $O_3$ concentrations to $NO_x$ and VOC emissions.

Although $SO_2$ concentrations are linearly related to $SO_2$ emissions, the chemical transport model

overestimates $SO_2$ concentrations and underestimates $SO_4^{2-}$ concentrations due to large uncertainties in
simulating the rapid conversion of $SO_2$ to $SO_4^{2-}$ during haze episodes (Zhang et al., 2019). To address this





deficiency, we adjusted the $SO_2$ emissions using the observed $SO_4^{2-}/SO_2$ ratio such that the RSM
predictions matched both the observed $SO_2$ and $SO_4^{2-}$ concentrations. Since $SO_4^{2-}$ concentrations are quite
linearly related to $SO_2$ emissions when $NH_3$ emissions are at moderate levels (Wang et al., 2011), we
assume that the unaccounted for $SO_2$-to-$SO_4^{2-}$ conversion pathway contributes to differences in the
observed and simulated $SO_4^{2-}/SO_2$ ratios. Under this assumption, simulated $SO_2$ concentrations are
overestimated by the same ratio (α) that secondary $SO_4^{2-}$ ($C_{s-SO4}^s$) concentrations are underestimated (see
E3 and E4). The primary $SO_4^{2-}$ concentration ($C_{p-SO4}^s$) was removed from the total $SO_4^{2-}$ concentration in
these calculations, because primary $SO_4^{2-}$ is directly emitted and not related to the conversion of $SO_2$ to
$SO_4^{2-}$ (see E4).
$$C_{SO2}^o = \frac{1}{\alpha} \times r_{SO2} \times C_{SO2}^s \qquad \text{(E3)}$$
$$C_{SO4}^o = \alpha \times r_{SO2} \times C_{s-SO4}^s + C_{p-SO4}^s \qquad \text{(E4)}$$
$$\alpha = \left( \frac{C_{SO2}^o}{C_{SO4}^o - C_{p-SO4}^s} \Big/ \frac{C_{SO2}^s}{C_{SO4}^s} \right)^{1/2} \qquad \text{(E5)}$$
The adjusted $SO_2$ emission ratio ($r_{SO2}$) is estimated by taking the ratio of observed $SO_2$ ($C_{SO2}^o$) to simulated
$SO_2$ ($C_{SO2}^s$) multiplied by α, which accounts for the model deficiency in simulating the rapid conversion
of $SO_2$ to $SO_4^{2-}$. For simplification, here we estimate the α value at a domain and temporal averaged level
(i.e., identical across the space and time), though such ratio might vary with time and space. The α is
smaller than 1 because the observed $SO_4^{2-}/SO_2$ is usually greater than the simulation. The inclusion of the
α may help the response model avoid the underestimation of $SO_2$ emissions.
Using the adjusted $NO_x$, VOC, and $SO_2$ emissions from previous steps, we next adjusted $NH_3$
emissions such that RSM predictions of $NO_3^-$ concentrations matched observations:
$$\Delta NO_3^- = (C_{NO3}^o - C_{NO3}^s) = RSM_{NO3}(r_{NOx}, r_{VOC}, r_{SO2}, r_{NH3}) \qquad \text{(E6)}$$
where $r_{NH3} = E'_{NH3}/E^*_{NH3}$, $E'_{NH3}$ is the adjusted $NH_3$ emissions, and $E^*_{NH3}$ is the prior $NH_3$ emissions.
After updating the emissions of the four gaseous precursors, the secondary portion of $PM_{2.5}$ was



correspondingly determined, including the secondary organic aerosol contributed by the VOC emissions.
Finally, the primary $PM_{2.5}$ emissions were adjusted to provide agreement between simulated and observed
total $PM_{2.5}$ concentrations:
$$\Delta PM_{2.5} = (C^o_{PM2.5} - C^s_{PM2.5}) = RSM_{PM2.5}(r_{NOx}, r_{VOC}, r_{SO2}, r_{NH3}, r_{p-PM2.5}) \quad (E7)$$
where $r_{p-PM2.5} = E'_{p-PM2.5}/E^*_{p-PM2.5}$, $E'_{p-PM2.5}$ is the adjusted primary $PM_{2.5}$ emissions, and $E^*_{p-PM2.5}$
is the prior primary $PM_{2.5}$ emissions.
The prior emissions used here were based on a bottom-up inventory developed for 2017. Since our
study focuses on periods in 2019 and 2020, we first use the response model to adjust the 2017 emission
inventory to match the two study periods. The first study period was defined as 1 January – 31 March
2019 to capture changes in activity due the Spring Festival.  The second study period was defined as the
same three months in 2020 to capture the COVID-19 shutdown in NCP, which overlapped the 2020 Spring
Festival holiday. We defined three sub-periods within the three months in each year as pre-shutdown
(Period 1), shutdown (Period 2), and post-shutdown (Period 3). The days selected for sub-periods differed
in 2019 and 2020 due to differences in the dates and lengths of the shutdowns. For 2019, we defined
Period 1: 1–29 Jan. (29 days); Period 2: 30 Jan. – 18 Feb. (20 days), which is a week before and after the
2019 Lunar New Year holidays; and Period 3: 19 Feb. – 31 Mar. (41 days).  For 2020, we defined Period
1: 1–22 Jan. (22 days); Period 2: 23 Jan. – 5 Mar. (33 days), which is from the date that Chinese authorities
began targeted transportation shutdowns until all human activities began recovering in early March
(http://www.gov.cn/index.htm); and Period 3: 6–31 Mar. (26 days).
The RSM was developed using ambient concentrations from simulations with the Community
Multiscale Air Quality (CMAQ, version 5.2.1) model, which incorporated meteorological fields from the
Weather Research and Forecasting (WRF, version 3.8) model. The WRF-CMAQ system was configured
as in our previous studies, and model performance for meteorological variables and pollutant
concentrations was evaluated (Ding et al., 2019). The RSM was developed following the same design as





our previous study (Xing et al., 2017), in which the polynomial response functions for $O_3$, $PM_{2.5}$ and $PM_{2.5}$
components were fitted by 40 brute-force CMAQ simulations. Specifically, deep-learning technology was
used to fit response surfaces for the three months in 2019 and 2020 using CMAQ simulations for baseline
and zero-out emissions conditions (Xing et al., under review) (see Figure 2). The response surfaces were
developed using year-specific meteorology based on WRF simulations to account for differences in
meteorological conditions between 2019 and 2020.

Measurements of ambient concentrations of $NO_2$, $SO_2$, $O_3$ and $PM_{2.5}$ were obtained from the China

National Environmental Monitoring Centre (http://106.37.208.233:20035/). Measurements of $PM_{2.5}$
chemical components, including $NO_3^-$ and $SO_4^{2-}$, were provided by the urban PM data analysis platform
in the 2+26 cities of Beijing-Tianjin-Hebei and surrounding regions (http://106.37.181.120:9011/bfs). All
monitoring data were given as hourly-averaged concentrations at the monitoring sites shown in Figure 2.
As in our previous RSM studies, daily daytime $O_3$ concentrations were analyzed based on afternoon
averages (12:00pm-6:00pm local time), and $PM_{2.5}$ concentrations were based on daily 24-hour averages
(Xing et al., 2018). Since the monitors sample pollutants at discrete locations and measurements are not
available for all days at all sites, provincial average concentrations were used to facilitate adjustments
domain-wide for all days in each study period. The provincial average concentrations were calculated
using spatially and temporally matched simulated and observed values.
**2.2 Hypothetical emissions without shutdown effects**

The actual emissions can be derived using observed concentrations and the response model.

However, hypothetical emissions under the assumption of no shutdown effects are also needed to estimate
the changes in emissions due to the 2019 and 2020 shutdowns. We estimate the hypothetical emissions
using the temporal profiles of sectoral emissions from the bottom-up inventory in combination with the
derived (actual) emissions for the pre- and post-shutdown periods. We assume that the Spring Festival
shutdowns in 2019 have negligible influence on emissions during the periods before and after the





shutdown (i.e., Period 1 and Period 3, respectively), while the COVID-19 pandemic in 2020 might have
had lag effects after the shutdown due to reduced economic activity or relaxed pollutant controls. However,
we concentrate our analysis of COVID-19 impacts on emissions and air quality in the official shutdown
period only (Period 2). The hypothetical no-shutdown emissions for Period 2 (noted as Period 2H) are
estimated using ratios of emissions for Period 2 and Period 1 and 3 based on the temporal profile of the
bottom-up inventory which only reflects the natural evolution of emissions across a year for each sector.
This approach develops hypothetical emissions following the typical variation in emissions without
shutdown effects. Note that we use the temporal profile to determine the change in Period 2 emissions
relative to Period 1 and 3, and so emissions from both Period 1 and 3 are needed to estimate Period 2H
emissions.

The emission changes due to the COVID-19 shutdown can be estimated by taking the difference of

emissions in Period 2, derived from the response model, and emissions in Period 2H, estimated from
emissions in Period 1 and 3 using the temporal profile of bottom-up sectoral emissions. The impacts of
emission changes during the COVID-19 shutdown on $PM_{2.5}$ and $O_3$ concentrations are then estimated with
the RSM. In addition to the combined impacts of emission changes from multiple species, we estimate the
impacts of individual pollutant emissions on $PM_{2.5}$ and $O_3$. Due to the nonlinearity of emission-
concentration response functions, the impacts of individual pollutant emissions can vary significantly
when other pollutant emissions are change simultaneously (Xing et al., 2018). To simplify the evaluation,
we define an incremental method for analyzing the individual pollutant impacts in this study by adding
incremental changes in pollutant emissions to the previous simulation in the following order: $NO_x$, VOC,
$NH_3$, $SO_2$ and primary $PM_{2.5}$, as described in Table 1. The impacts of individual pollutant emissions on
$O_3$ and $PM_{2.5}$ concentrations are then estimated from the difference between the incrementally adjusted
simulation and the previous one. Note that this approach is an approximation, and the impacts of individual
pollutants could change if a different order is used.



## 3. Results

**3.1 Emission changes due to the shutdown**

Using the response model, the daily emissions of $NO_x$, VOC, $NH_3$, $SO_2$ and primary $PM_{2.5}$ in NCP are estimated for three periods in 2019 and 2020, as summarized in Figure 3 and detailed in Table 2 by provinces.

For Period 1 before the activity disruptions, the emissions of $NO_x$, $SO_2$, and VOC in NCP decreased by 11%, 25%, and 8% between 2019 and 2020, respectively. These reductions reflect the progress of air pollution controls between 2019 and 2020, and demonstrate the ability of the model to capture emission changes from routine air pollution control actions. The p-$PM_{2.5}$ emissions also significantly decreased in Beijing-Tianjin-Hebei provinces but increased in Shandong and Henan. The $NH_3$ emissions did not change during this two-year period, since $NH_3$ is not considered in current policies.

Activity reductions occurred in Period 2 in both 2019 and 2020, although the shutdown due the Spring Festival in 2019 is much shorter than the COVID-19 shutdown in 2020. The emissions of $NO_x$, $SO_2$ and p-$PM_{2.5}$ in Period 2 in 2020 are substantially lower than in 2019 (29%, 22% and 73%, respectively). The decreases of $NO_x$ and p-$PM_{2.5}$ for Period 2 between 2019 and 2020 are larger than the decreases for Period 1, which did not experience shutdowns. Such results suggest that the COVID-19 shutdown in 2020 had longer and stronger impacts on emissions than the Spring Festival shutdown in 2019. Interestingly, emissions of $NH_3$ and VOC increased significantly (by 5% and 14%) from 2019 to 2020 in Period 2. These changes are likely due to the temporal variations of emissions of both species, which are enhanced in warmer months due to stronger evaporation. Period 2 in 2020 extended farther into the Spring (until early March) than Period 2 in 2019, and thus led to increased evaporative emissions of $NH_3$ and VOC. These results also demonstrate the importance of developing emissions with high temporal resolution.

For Period 3 after the shutdown, the decreases of $NO_x$ emissions (14%) are similar to those in Period


1 (11%), indicating the recovery of the activity. However, the emissions of VOC and p-$PM_{2.5}$ are much
lower in Period 3 in 2020 compared to that in 2019, suggesting the lag effects after the COVID-19
shutdown in 2020. In contrast, the small increases of $SO_2$ emissions in 2020 (2%) might be associated
with the extended central heating activity through the end of March in 2020, compared with mid-March
in 2019. Higher $NH_3$ emissions in Period 3 in 2020 than 2019 are also due to the larger coverage of warm
days in Period 3 of 2020. $NH_3$ emissions show the strongest monthly variations among all pollutants
(Figure 3). Similarly, increases in VOC emissions are also driven by the change of meteorological
conditions (i.e., the higher air temperature in March leads to a larger evaporative emissions), though the
growth of VOC emissions from Period 1 to Period 3 is reduced by the COVID-19 shutdown in 2020. Such
results also demonstrate that the response model can capture the temporal variations of emissions even in
cases where emissions are strongly coupled with meteorological conditions.

The influence of the shutdown is estimated as the difference in emissions between Period 2H

(hypothetical emissions without shutdown effects) and Period 2 (actual emissions), as shown in Figure 3
(grey and red bars respectively) and detailed in Table 3 by NCP province. Due to the COVID-19 shutdown
in 2020, emissions of $NO_x$, VOC and $PM_{2.5}$ decreased substantially by 51%, 67% and 63%, respectively.
$SO_2$ emissions also decreased by 28%, while $NH_3$ emissions experienced very small increases (+2%)
which might be associated with increased activities in rural areas (e.g., potential $NH_3$ emission sources
like stool burning) as many people relocated from megacities to small towns or the countryside. Compared
to the effects of the Spring Festival in 2019, the COVID-19 shutdown led to greater reductions in $NO_x$,
$SO_2$ and $PM_{2.5}$ emissions. The smaller VOC reduction in 2020 compared to 2019 might be due to the
difference in temporal coverage of Period 2 in the two years (i.e., there were more warm days in Period 2
in 2020). Note that the hypothetical emissions in Period 2H are estimated based on the assumption of no
shutdown effects in both Period 1 and Period 3. Therefore the reduction of those pollutant emissions in
2020 might be even larger considering the lag effects of COVID-19.





### 3.2 The shutdown effects on ambient concentrations

Using the RSM, we predicted concentrations based on the updated emissions from the response-based inversion model. In general, the simulated concentrations based on the adjusted emissions matched well with the observed concentrations, as shown in Figure 4 for NCP averages and detailed by province in Figure S1-12. More important, during the shutdown period in both years, the simualtions using adjusted emissions without considering shutdown influences significantly overestimate the $NO_2$ concentations in 2019 and 2020 by 61% and 81%, respectively. The high-biases in 2019 and 2020 are reduced to within 1% in the simualtion with consideration of shutdown effects (Figure 4a).

The results for $O_3$ are quite interesting, as simulated $O_3$ concentrations are close to observations in both simulations with and without consideration of shutdown influences (Figure 4b). The apparent insenstivity of $O_3$ concentrations to emission changes during the shutdown can be explained by the nonlinear response of $O_3$ to its two percurors, $NO_x$ and VOC. In Figure 5a, we compare the response of $O_3$ concentrations for two $NO_x$ and VOC emission change pathways starting from the hypothetical emissions for no-shutdown conditions (black symbol in Figure 5a). Since $NO_x$ emissions clearly decreased due to the shutdown, the $O_3$ concentrations would increase if VOC emissions remained constant (following the green line to the green symbol in Figure 5a). Yet the simulation without consideration of VOC emission changes would result in a high bias of simulated $O_3$ concentrations compared to the observations by 49% in 2019 and 29% in 2020. The low observed $O_3$ concentrations during Period 2 in both years indicates that VOC emission reductions must have occurred to maintain the suppressed $O_3$ level (following the red line to the red symbol in Figure 5a). Consistent with this interpretation, the simulated $O_3$ concentrations agree well with observations (e.g., normarlized mean bias, NMB < 3%) when both $NO_x$ and VOC emission reductions are represented.

The substantial reduction of $NO_x$ emissions also resulted in noticable decreases in $NO_3^-$ concentrations (black and green lines in Figure 4c). However, the low bias in $NO_3^-$ predictions cannot be



readily mitigated by adjusting the $NH_3$ emissions, because the substantial decreases in $NO_x$ emissions
associated with the shutdown result in a strong $NH_3$-rich conditions, where $NO_3^-$ concentrations are less
sensitive to $NH_3$ emissions increases. The response of $NO_3^-$ concentrations to pathways of $NO_x$ and $NH_3$
emission changes is depicted in Figure 5b ($SO_2$ and VOC emissions are also changing simutaneously with
$NO_x$). A larger decrease in simulated than observed $NO_3^-$ concentrations is associated with the $NO_x$
emission reductions, but the change of $NH_3$ emissions can hardly increase the $NO_3^-$ concentrations under
such strong $NH_3$-rich conditions. Therefore, the model predicted no $NH_3$ changes in 2019, but very small
increases of $NH_3$ emissions (+2%) in 2020 due to the increased activities in rural areas which slightly
reduced the $NO_3^-$ low biases (NMB from -12% to -11%).

The large reduction in $SO_2$ emissions estimated with the response model during the 2020 shutdown

considerably reduced the high biases in simulated $SO_2$ and $SO_4^{2-}$ concentrations (Figure 4d-f). However,
the $SO_4^{2-}$ biases are still considerable after the emission adjustment because a large fraction of $SO_4^{2-}$ might
come from primary sources, which need further investigation especially for its contribution to p-$PM_{2.5}$.

Agreement between the simulated and observed $PM_{2.5}$ concentrations also improves when

accounting for the reductions in primary $PM_{2.5}$ emissions estimated with the response model in both years
(Figure 4g). Another interesting finding is that the simulated $PM_{2.5}$ concentrations with consideration of
all emission changes due to the shutdown (red line in Figure 4g) are quite similar to $PM_{2.5}$ predictions
without consideration of the shutdown impacts (black line in Figure 4g). The same behavior is evident for
$O_3$ concentrations (red and black lines in Figure 4b). As discussed above, the reductions in emissions of
multiple species during the shutdown had compensating influences on air quality, and the overall effects
of the emission changes on $O_3$ and $PM_{2.5}$ concentrations were neutralized to a relatively small level.
**3.3 Impacts of individual emission changes from the shutdown on $O_3$ and $PM_{2.5}$ concentrations**

To further investigate the individual impacts of emission changes of each pollutant on $O_3$ and $PM_{2.5}$

concentrations, we conducted sensitivity analysis by sequentially adding each incremental emission



change into the model system and then calculating the associated changes in $O_3$ and $PM_{2.5}$ concentrations.
By incrementally adding the impacts of emission changes of five pollutants ($\Delta NO_x$, $\Delta VOC$, $\Delta NH_3$, $\Delta SO_2$,
and $\Delta$p-$PM_{2.5}$), the concentrations change from the original simulation, without consideration of shutdown
impacts (noted as oSIM, shown as grey bar in Figure 6), and ultimately reaching the observed levels (noted
as OBS, shown as narrow blue bars in Figure 6).
For $O_3$, the reduction of $NO_x$ emissions lead to a significant enhancement of $O_3$ (see $\Delta NO_x$) due to
the VOC-limited regime in winter (Xing et al., 2019), while such $O_3$ enhancement has been largely or
completely mitigated thanks to the simultaneous reduction of VOC emissions (see $\Delta VOC$) in both 2019
and 2020. This behavior is particularly evident in Henan and Shandong provinces which experienced
substantial VOC reductions during the shutdown (Table 3). Such benefits from simultaneous VOC
controls also occurred for $PM_{2.5}$ concentrations. Compared with $O_3$, the changes in $PM_{2.5}$ concentrations
are more complex to interpret due to the influence of emission changes for $SO_2$ ($\Delta SO_2$), $NH_3$ ($\Delta NH_3$) and
p-$PM_{2.5}$ ($\Delta$p-$PM_{2.5}$) in addition to $NO_x$ and VOC. Results suggest that the reductions of p-$PM_{2.5}$ emissions
tended to favor $PM_{2.5}$ decreases while the $\Delta SO_2$ and $\Delta NH_3$ emission changes have negligible influence.
Overall, reductions in p-$PM_{2.5}$ and VOC emissions helped mitigate potential $PM_{2.5}$ concentration
enhancements in most NCP provinces. Similar findings are suggested in Hang et al. (2020), which
observed enhanced secondary pollution during the COVID-19 period. The air quality impacts from the
unexpected controls during the COVID-19 shutdown suggest that strengthened controls on p-$PM_{2.5}$
emissions and well-balanced reductions in $NO_x$ and VOC emissions would be an effective strategy for
further improving air quality in NCP (Xing et al., 2018).

## 4. Summary and Conclusion

In summary, this study developed a response-based inversion modeling framework and applied it to
characterize the emission changes and associated air quality impacts during the 2019 Spring Festival and
the 2020 COVID-19 pandemic shutdown. Our results indicate that the response model can effectively



adjust the assumed prior emissions such that air quality predictions match well with observed
concentrations. The model also captures the temporal variations of emissions associated with changes in
meteorological conditions. The model may suffer some uncertainties from deficiencies in model chemical
mechanisms (e.g., conversion of $SO_2$ to $SO_4^{2-}$), as well as the quality of prior emissions and limited
coverage of observations. Difficulties are also found in estimating the $NH_3$ emission changes under strong
$NH_3$-rich conditions by using the current inversion method based on the concentration of PM chemical
components. However, with the continued growth in observational datasets from both surface monitors
and satellite retrievals, improvements in knowledge of atmospheric science, and development of advanced
assimilation technologies, the new response-based inversion model has great potential to further improve
the accuracy and efficiency of emission inventory updates. The importance of reliable bottom-up
inventories for defining prior emissions by sector, combined with the ability of the top-down inversion
model to rapidly adjust emissions for consistency with observations, demonstrates how bottom-up and
top-down emissions modeling methods are complementary.
The response model was applied in investigating the emission changes during the COVID-19
shutdown. The emission changes were estimated by comparing emissions for actual conditions with
emissions for hypothetical conditions assuming that the shutdown did not occur. Emission levels during
the COVID-19 shutdown period were estimated by applying the temporal profiles of sectoral emissions
from the bottom-up inventory. These estimates may suffer some uncertainties associated with the temporal
profiles and the assumption of no shutdown impacts during the post-shutdown period. Our results suggest
that the shutdowns in 2019 and 2020 had considerable impacts on air pollutant emissions. Longer and
stronger impacts are found in 2020 due to the COVID-19 pandemic compared to the Spring Festival of
the previous year. The anthropogenic emissions of $NO_2$, $SO_2$, VOC, and primary $PM_{2.5}$ in NCP were
reduced by 51%, 28%, 67% and 63%, respectively, due to the COVID-19 shutdown in 2020. The estimated
ratio might be slightly underestimated considering the lag effects after the COVID-19 shutdown. We also



found that emission changes associated with the shutdown periods had limited impacts on surface $O_3$ and
$PM_{2.5}$ concentrations due to compensating effects of emission changes in different pollutants. Based on
our analysis, careful controls on $NO_x$ emission sources in NCP are recommended in combination with
simultanous controls on VOC and $NH_3$ sources. Such a comprehensive strategy would minimize the
potential negative impacts on air quality of $NO_x$ emission reductions during VOC-limited conditions in
winter. This study also illustrates that air quality improvements do not necessary follow from precursor
emission reductions, and multi-pollutant nonlinear response models are therefore critical tools for
representing the nonlinear relationship between emissions and concentrations in designing effective
control strategies.

## Data and code availability

The original data and code used in this study are available upon request from the corresponding
authors.

## Author contribution

JX & SL designed the methodology, conducted the analysis, and wrote the original draft. YJ
conducted the WRF-CMAQ simulation. SW & DD & ZD & JH helped with the bottom-up emission
inventory. YZ helped with the RSM model. All authors contribute to writing the paper.

## Acknowledgements

This work was supported in part by National Key R & D program of China (2018YFC0213805),
and National Natural Science Foundation of China (21625701, 41907190). This work was completed on
the "Explorer 100" cluster system of Tsinghua National Laboratory for Information Science and
Technology. We thank Drs. Carey Jang, James Kelly, Jian Gao, and Jingnan Hu for contributions to the
study. The authors gratefully acknowledge the free availability and use of observation datasets.



## 374 Competing interests

The authors declare no competing financial interests.



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



**Table 1** Sensitivity analysis for quantifying the impacts of individual pollutant emission changes on air
quality

| No. | Emission | Objective | Noted |
|---|---|---|---|
| Sim-1 | All pollutants are used as the hypothetical emissions of Period 2H | To estimate the hypothetical concentrations without COVID impacts | oSIM |
| Sim-2 | Same as Sim-1 except $NO_x$ emissions are updated to actual emissions in Period 2 | To estimate the impacts of $NO_x$ emission changes on $O_3$ and $PM_{2.5}$ based on the difference between Sim-2 and Sim-1 | $\Delta NO_x$ |
| Sim-3 | Same as Sim-2 except VOC emissions are updated to actual emissions in Period 2 | To estimate the impacts of VOC emission changes on $O_3$ and $PM_{2.5}$ based on the difference between Sim-3 and Sim-2 | $\Delta VOC$ |
| Sim-4 | Same as Sim-3 except $NH_3$ emissions are updated to actual emissions in Period 2 | To estimate the impacts of $NH_3$ emission changes on $PM_{2.5}$ based on the difference between Sim-4 and Sim-3 | $\Delta NH_3$ |
| Sim-5 | Same as Sim-4 except $SO_2$ emissions are updated to actual emissions in Period 2 | To estimate the impacts of $SO_2$ emission changes on $PM_{2.5}$ based on the difference between Sim-5 and Sim-4 | $\Delta SO_2$ |
| Sim-6 | Same as Sim-5 except primary $PM_{2.5}$ emissions are updated to actual emissions in Period 2 | To estimate the impacts of primary $PM_{2.5}$ emission changes on $PM_{2.5}$ based on the difference between Sim-6 and Sim-5 | $\Delta p\text{-}PM_{2.5}$ |







**Table 2** Daily emissions of five pollutants in NCP provinces based on the response model (unit: kt/day)

| 2019 | Period 1 (29 days, Jan 1 to Jan 29) | | | | | Period 2 (20 days, Jan 30 to Feb 18) | | | | | Period 3 (41 days, Feb 19 to Mar 31) | | | | |
|---|---|---|---|---|---|---|---|---|---|---|---|---|---|---|---|
| | $NO_x$ | $SO_2$ | $NH_3$ | VOC | p-PM$_{2.5}$ | $NO_x$ | $SO_2$ | $NH_3$ | VOC | p-PM$_{2.5}$ | $NO_x$ | $SO_2$ | $NH_3$ | VOC | p-PM$_{2.5}$ |
| Beijing | 0.49 | 0.07 | 0.20 | 0.69 | 0.12 | 0.26 | 0.05 | 0.19 | 0.20 | 0.01 | 0.48 | 0.05 | 0.23 | 0.94 | 0.16 |
| Tianjin | 0.65 | 0.17 | 0.15 | 0.92 | 0.05 | 0.42 | 0.17 | 0.15 | 0.24 | 0.04 | 0.79 | 0.21 | 0.25 | 1.37 | 0.15 |
| Hebei | 5.64 | 2.01 | 1.18 | 3.67 | 1.97 | 3.47 | 1.62 | 1.27 | 1.43 | 1.51 | 5.95 | 1.90 | 2.77 | 6.26 | 1.92 |
| Shandong | 7.35 | 3.21 | 1.34 | 8.58 | 0.76 | 4.45 | 2.88 | 1.52 | 2.41 | 0.88 | 6.90 | 3.45 | 3.54 | 9.59 | 1.19 |
| Henan | 5.34 | 1.49 | 1.31 | 4.08 | 1.54 | 3.04 | 1.31 | 1.74 | 0.71 | 1.84 | 4.46 | 1.84 | 4.27 | 4.46 | 1.33 |
| NCP | 19.47 | 6.96 | 4.17 | 17.94 | 4.43 | 11.65 | 6.03 | 4.87 | 5.00 | 4.28 | 18.58 | 7.45 | 11.07 | 22.62 | 4.76 |



| 2020 | Period 1 (22 days, Jan 1 to Jan 22) | | | | | Period 2 (33 days, Jan 23 to Mar 5) | | | | | Period 3 (26 days, Mar 6 to Mar 31) | | | | |
|---|---|---|---|---|---|---|---|---|---|---|---|---|---|---|---|
| | $NO_x$ | $SO_2$ | $NH_3$ | VOC | p-PM$_{2.5}$ | $NO_x$ | $SO_2$ | $NH_3$ | VOC | p-PM$_{2.5}$ | $NO_x$ | $SO_2$ | $NH_3$ | VOC | p-PM$_{2.5}$ |
| Beijing | 0.38 | 0.04 | 0.20 | 0.65 | 0.01 | 0.23 | 0.03 | 0.20 | 0.27 | 0.01 | 0.28 | 0.04 | 0.24 | 0.70 | 0.09 |
| Tianjin | 0.64 | 0.12 | 0.15 | 0.87 | 0.02 | 0.44 | 0.12 | 0.17 | 0.44 | 0.03 | 0.71 | 0.18 | 0.30 | 1.20 | 0.10 |
| Hebei | 5.28 | 1.34 | 1.18 | 3.12 | 1.73 | 3.15 | 1.16 | 1.54 | 1.92 | 0.81 | 4.97 | 1.67 | 3.49 | 4.72 | 0.75 |
| Shandong | 6.57 | 2.55 | 1.34 | 8.02 | 0.85 | 3.28 | 2.25 | 1.88 | 2.44 | 0.16 | 5.87 | 3.57 | 4.52 | 8.44 | 0.14 |
| Henan | 4.50 | 1.15 | 1.31 | 3.84 | 2.26 | 1.13 | 1.14 | 1.31 | 0.64 | 0.16 | 4.09 | 2.13 | 5.49 | 3.13 | 0.10 |
| NCP | 17.37 | 5.19 | 4.17 | 16.51 | 4.88 | 8.23 | 4.69 | 5.10 | 5.71 | 1.17 | 15.93 | 7.59 | 14.03 | 18.18 | 1.19 |
| Δ2020-2019 | -11% | -25% | 0% | -8% | 10% | -29% | -22% | 5% | 14% | -73% | -14% | 2% | 27% | -20% | -75% |


(p-PM$_{2.5}$ = primary PM$_{2.5}$)





**Table 3** The shutdown-impacts on the emission of five pollutants in NCP provinces

| 2019 | NO$_x$ | | SO$_2$ | | NH$_3$ | | VOC | | p-PM$_{2.5}$ | |
|---|---|---|---|---|---|---|---|---|---|---|
| | kt/Day | % | kt/Day | % | kt/Day | % | kt/Day | % | kt/Day | % |
| Beijing | -0.23 | -47% | -0.01 | -21% | 0.00 | 0% | -0.56 | -73% | -0.15 | -93% |
| Tianjin | -0.30 | -41% | -0.02 | -10% | 0.00 | 0% | -0.95 | -80% | -0.07 | -62% |
| Hebei | -2.33 | -40% | -0.34 | -17% | 0.00 | 0% | -3.54 | -71% | -0.51 | -25% |
| Shandong | -2.67 | -37% | -0.46 | -14% | 0.00 | 0% | -6.78 | -74% | -0.10 | -10% |
| Henan | -1.85 | -38% | -0.48 | -27% | 0.00 | 0% | -3.39 | -83% | 0.39 | 27% |
| NCP | -7.38 | -39% | -1.31 | -18% | 0.00 | 0% | -15.23 | -75% | -0.43 | -9% |


| 2020 | NO$_x$ | | SO$_2$ | | NH$_3$ | | VOC | | p-PM$_{2.5}$ | |
|---|---|---|---|---|---|---|---|---|---|---|
| | kt/Day | % | kt/Day | % | kt/Day | % | kt/Day | % | kt/Day | % |
| Beijing | -0.10 | -30% | -0.01 | -18% | 0.00 | 2% | -0.39 | -59% | -0.07 | -85% |
| Tianjin | -0.24 | -35% | -0.03 | -18% | 0.00 | 2% | -0.60 | -58% | -0.04 | -59% |
| Hebei | -1.98 | -39% | -0.31 | -21% | 0.03 | 2% | -1.89 | -50% | -0.43 | -35% |
| Shandong | -2.95 | -47% | -0.75 | -25% | 0.04 | 2% | -5.80 | -70% | -0.31 | -66% |
| Henan | -3.16 | -74% | -0.76 | -40% | 0.03 | 2% | -3.10 | -83% | -1.10 | -87% |
| NCP | -8.42 | -51% | -1.85 | -28% | 0.10 | 2% | -11.77 | -67% | -1.95 | -63% |


(p-PM$_{2.5}$ = primary PM$_{2.5}$)



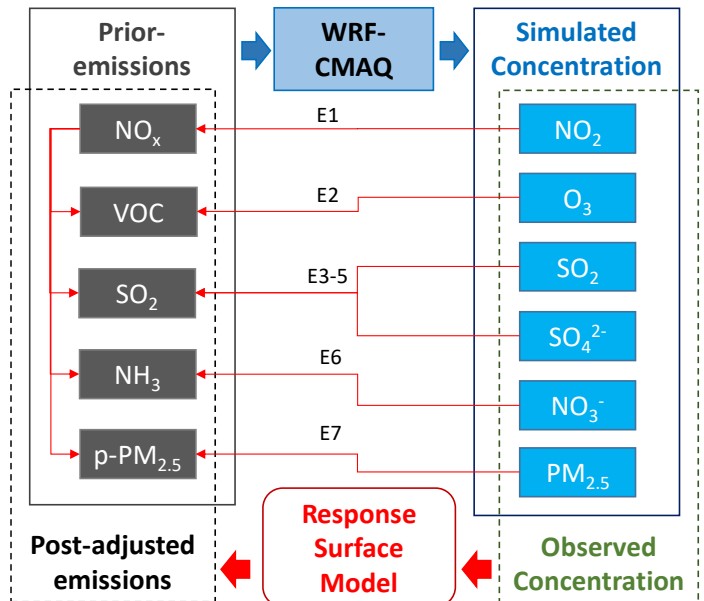


**Figure 1** The response modeling framework for adjusting the emissions (the E1-7 are equations used to
adjusted emissions, which are detailed in the text)



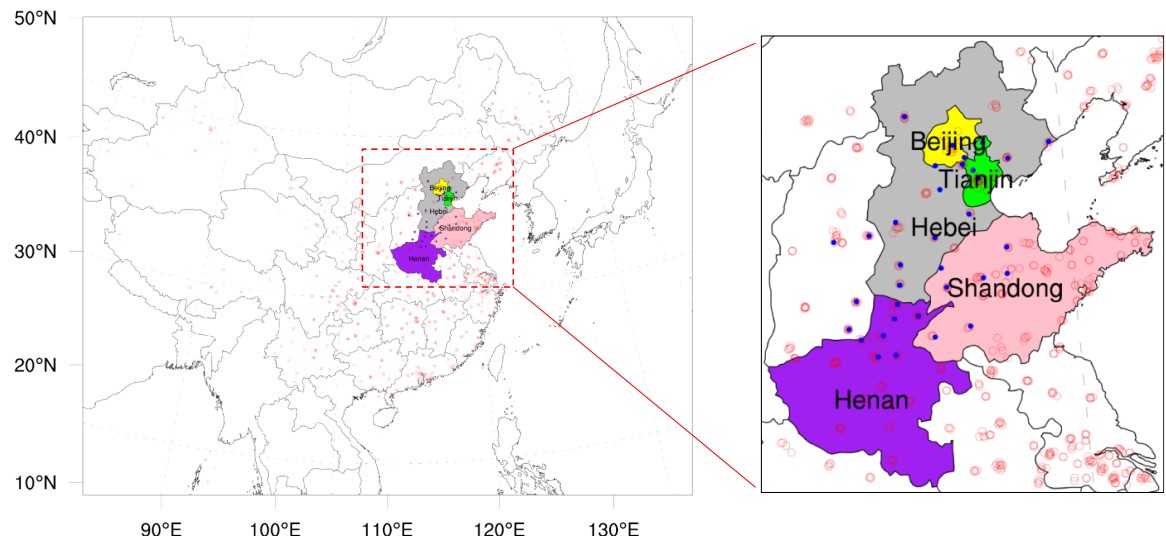

**Figure 2** Simulation domain and observation sites in five provinces of North China Plain (red dots: surface monitor sites for NO₂, SO₂, O₃ and PM₂.₅; blue dots: monitor sites for PM₂.₅ chemical compoments)





468

469

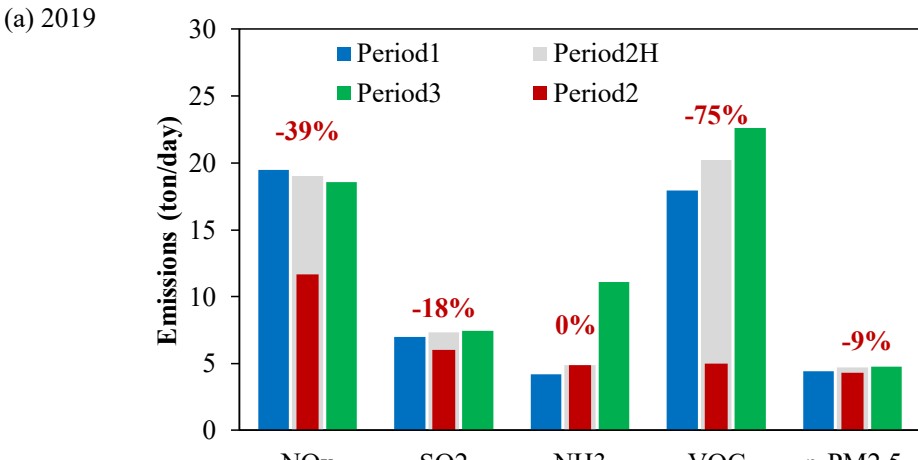

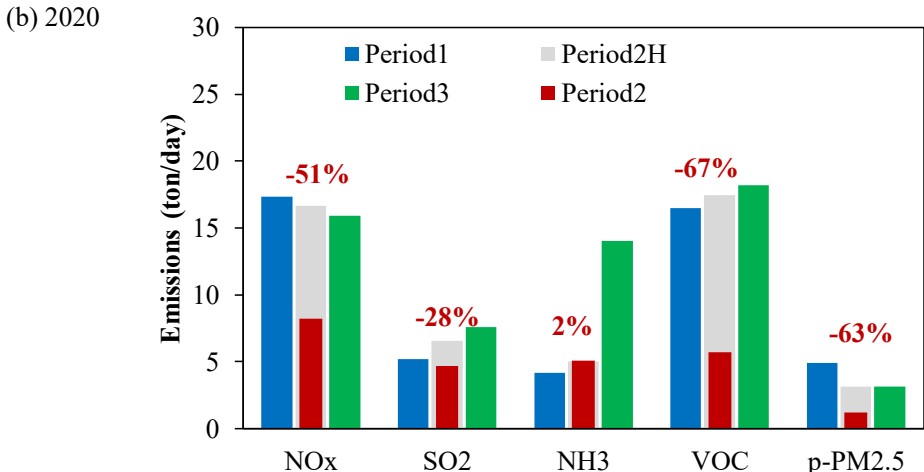

**Figure 3** Daily emissions during pre-shutdown (Period 1, blue), shutdown (Period 2, red), and post-shutdown (Period 3, green) periods in 2019 and 2020. Period 2H (grey) is the hypothetical emissions without reduced activity during the 2019 holiday or 2020 COVID-19 shutdown; the red number indicates the percent change in emissions due to the shutdown in Period 2.

474



**Figure 4** Comparison of the simulated and observed average concentrations in NCP (the percentage numbers indicate the normalized mean biases in hypothesis and actual simulations respectively for Period 2. Blue dots: observations; Black line: simualtions using adjusted emission with no consideration of shutdown influences; Red line: simualtions using adjusted emission with consideration of shutdown influences; Green line: simualtions using adjusted emission with consideration of shut-down influences without VOC for $O_3$, $NH_3$ for $NO_3^-$, $SO_2$ for $SO_4^{2-}$, primary $PM_{2.5}$ for $PM_{2.5}$; unit: μg m$^{-3}$)



481

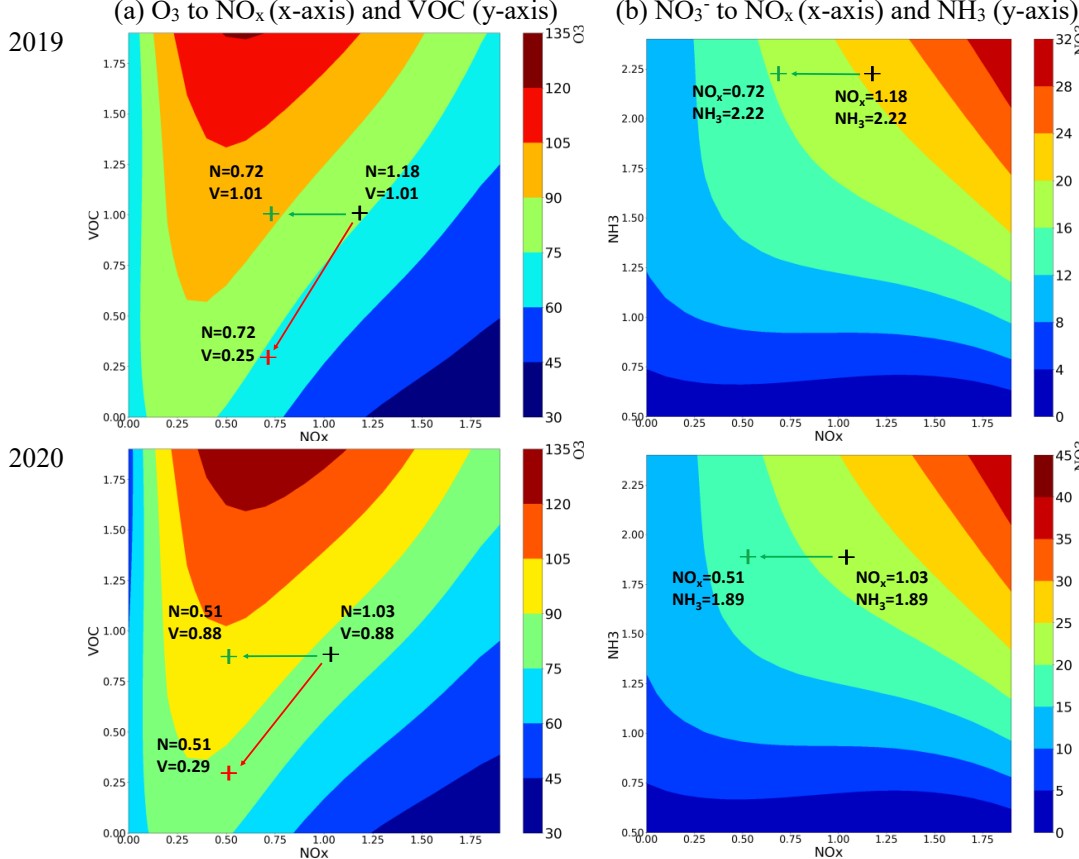

482

**Figure 5** Implication of emission changes from the O₃ and NO₃⁻ response isopleths during shutdowns (the axes indicate emission ratios relative to the prior emissions; black symbol: adjusted emission ratios with no consideration of shutdown; red symbol: adjusted emission ratios with consideration of shutdown; green symbol: adjusted emission ratios without considering simutanous VOC changes for O₃, and NH₃ changes for NO₃; background color: O₃ and NO₃⁻ concentrations, μg m⁻³)

488



489

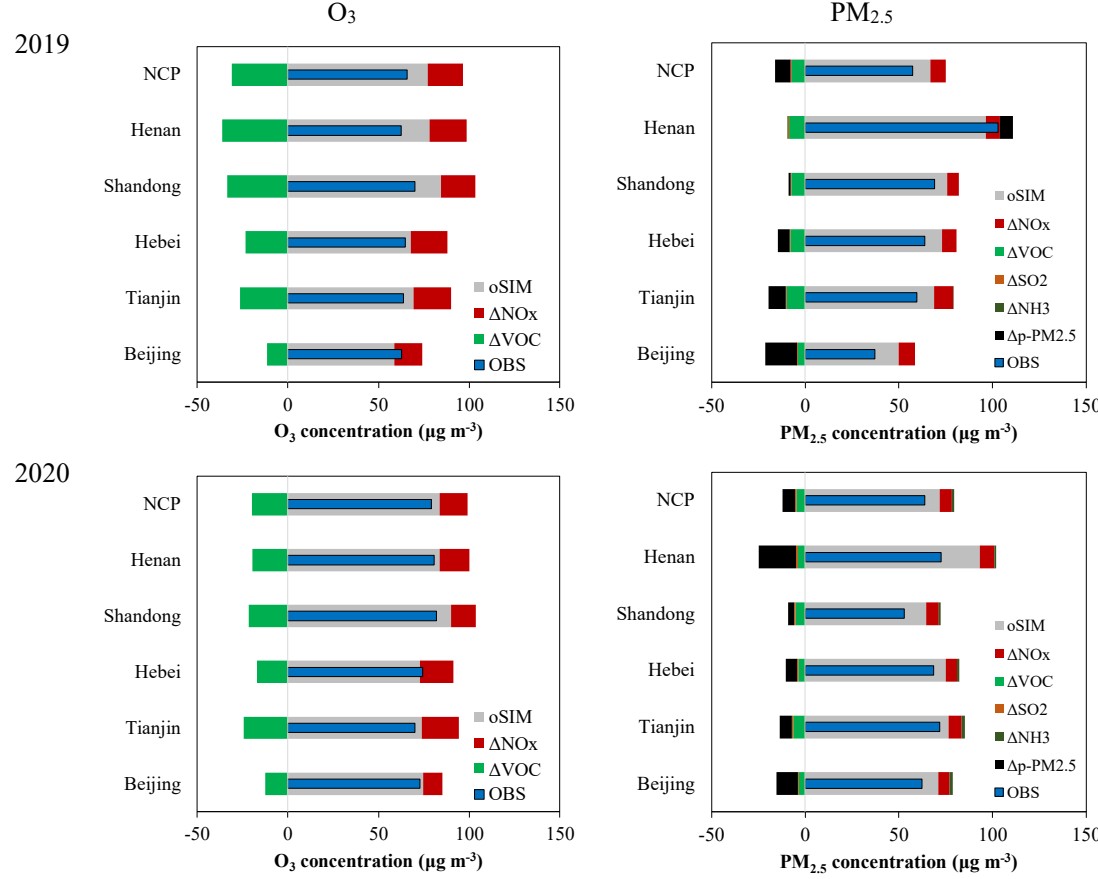

490

**Figure 6** Contributions to the changes of $O_3$ and $PM_{2.5}$ concentrations during Period-2 (OBS: observation; oSIM: no consideration of shutdown; $\Delta NO_x$: impacts due to the change of $NO_x$ emissions; $\Delta VOC$: impacts due to the change of VOC emissions; $\Delta NH_3$: impacts due to the change of $NH_3$ emissions; $\Delta SO_2$: impacts due to the change of $SO_2$ emissions; $\Delta p$-$PM_{2.5}$: impacts due to the change of primary $PM_{2.5}$ emissions)