# Peer review of "Quantifying the emission changes and associated air quality impacts during"

_Atmospheric Chemistry and Physics, 2020_

## Referee Comment (RC1) · Anonymous Referee #1 · 26 Aug 2020

The manuscript titled "Quantifying the emission changes and associated air quality impacts during the COVID-19 pandemic in North China Plain: a response modeling study" by Xing et al. quantifies emission changes during the shutdown in spring 2020 caused by the Covid-19 restrictions in the north China plane. The emission changes in 2020 are estimated from observed concentrations using response-based inversion and are compared to conditions in 2019 and hypothetical conditions. The overall scientific question addresses an interesting and up-to-date issue which is relevant for air quality research and analysis.

The used response-based inversion model ("response model") includes a response

surface model ("RSM") developed in previous studies (e.g. Xing et al. 2018) which provides emission-concentration relations. Based on this, emissions of five pollutants (NOx, VOC, SO2, NH3, PM2.5) are corrected with respect to locally observed concentrations. The chosen approach appears to induce suitable corrections of emissions, however some points might need to be clarified/adopted:

1. Talking about emission-inversion, a more detailed description of existing top-down inversion methods is needed in the introduction which demonstrates the novelty of the presented method more clearly (ll. 52). Methods for emission optimization in the context of inverse modeling of parameters and chemical data assimilation should be noted and shortly discussed with respect to advantages and disadvantages of the new method.

2. What remains somehow unclear is the way how emissions are changed by the response model. Are emissions only corrected locally (i.e. the emissions at the location of the observations) or does the response model consider inverse non-local transport and transformation processes (i.e. correction of emissions at remote locations)? What is the temporal extension of the corrections? Maybe, this can be described more clearly in the manuscript. In case of non-local corrections, it would be interesting to show the spatial patterns of corrections induced by the inversion.

3. Concerning the evaluation with observations: In data assimilation, forecasts are usually evaluated by independent observations, which are not considered in the optimization procedure. This is the standard way to investigate the usefulness of applied corrections. The corrected forecasts should fit the observations used for correction for any consistent method by definition. Thus, an evaluation with those observations does not provide additional information in the methodical point of view.

Smaller comments:

- In the description of the correction of SO2-emissions: Are primary SO4 ($C^s_{p-SO4}$, ll. 118, Eq. (E4) ) concentrations assumed to be correct? This might be worth mentioning in the description.

- Maybe related to point 2 (above): How does are the emissions from the bottom-up inventory of 2017 corrected by the response model? Is this also based on the observations used for correction later on? Or are the total annual emissions scaled by some correction values (i.e. constant correction of emissions keeping the annual and diurnal variations constant)?

- It might be worth explaining the estimation of the hypothetical no-shutdown emissions a bit more in detail (ll. 186). Are these estimated from temporally- and spatially averaged ratios between the periods?

- As far as I understand, Figure 4 shows averaged observations over the entire plain. Such spatially-averages should be used with caution as observations from single stations might be missing for some time (as noted in ll. 172). This hampers the interpretation of the temporal evolution of the plotted observations. Moreover, it needs to be made clear if the simulated concentrations shown in Fig.4 only refer to these stations, which provide used observations at each time. Drawing a continuous line might be misleading in case the simulated concentrations include different stations at different times.

Technical comments:

- The reference to Figure 2 in line 162 is not clear. It seams to be not connected to Fig. 2 of this manuscript. What is the content of the cited manuscript by "Xing et al, under review"?

[Figure]

- The x-axis of Figure 4 is not defined. Does this refer to the day of year?

- Fig. 4: The red line is hardly visible e.g. in Figure 4c. It might be useful to plot it as somehow thicker/ dashed line.

- Fig. 6: If I did not miss it, it would be interesting to include the concentrations resulting from changes in all emissions due to the shutdown (Period 2, incl. Shutdown-effects) in Figure 6 (maybe instead of plotting observations). This would make the overall changes in concentrations due to the shutdown more clear than comparing the simulated no-shutdown concentrations with observations. If this comparison was made for a specific purpose, maybe did not became clear to me in the text.

- Line 266: The response to O3 to NOx and VOC appears to be quite linear in the local regime as shown in Fig. 5. I would suggest to replace the formulation in ll. 266 by e.g. "opposite response" or "compensating effects".

- Line 285: To which decrease in NO3 concentrations is here refereed to? Is this the decrease in Period 2 compared to Period 1?

- Finally, three small technical suggestions (i) delete the "a" in line 280: "... result in strong NH3-rich conditions, ...", (ii) add an "a" in line 302: "..., we conducted a sensitivity analysis ..." and (iii) in line 340: e.g. "... was applied to the investigation of emission changes ..."

---

## Referee Comment (RC2) · Anonymous Referee #2 · 27 Aug 2020

Xing et al. used the response surface model to estimate the emission changes based on the air pollutants concentration changes during COVID-19 in China. Accurate and timely estimate of emission changes are critical to investigate how the air pollutants response to rapid environment changes, such as halt of transportation, slowdown of industry and energy sector during COVID-19, which are missing in recently published journal articles studying the air quality response to COVID-19. The methodology proposed in this study provides a promising framework connect real-time emission changes with abrupt environment changes. I am also very satisfied when the authors provide hypothetical individual emission changes on the influence of ambient concentration changes (section 3.3), which is very helpful to design the multi-pollutants control

strategies in China. The manuscript fits for the journal as well, and I suggest acceptance for this journal.

Minor comments: 1. L162: Fig 2 is not related to the reference pointed here; Also by looking at Fig. 2, there are more observations sites besides NCP. So I suggest the author rewrite the legend for Fig. 2

Figures Fig 3. Consider to put subscript letter for those air pollutants

Fig 4. Consider to put the simulations with the prior emission (without using the RSM to adjust) for comparisons purpose.
* * *

---

## Author Comment (AC1) · 3 Oct 2020

[Comment]: The manuscript titled "Quantifying the emission changes and associated air quality impacts during the COVID-19 pandemic in North China Plain: a response modeling study" by Xing et al. quantifies emission changes during the shutdown in spring 2020 caused by the Covid-19 restrictions in the north China plane. The emission changes in 2020 are estimated from observed concentrations using response-based inversion and are compared to conditions in 2019 and hypothetical conditions. The overall scientific question addresses an interesting and up-to-date issue which is relevant for air quality research and analysis. The used response-based inversion model

("response model") includes a response surface model ("RSM") developed in previous studies (e.g. Xing et al. 2018) which provides emission-concentration relations. Based on this, emissions of five pollutants (NOx, VOC, SO2, NH3, PM2.5) are corrected with respect to locally observed concentrations. The chosen approach appears to induce suitable corrections of emissions, however some points might need to be clarified/adopted.

[Response]: We thank the reviewer for recognition of the implications of the results of the analysis presented, and overall positive comments. We have followed all the comments and revised manuscript accordingly.

[Comment]: 1. Talking about emission-inversion, a more detailed description of existing topdown inversion methods is needed in the introduction which demonstrates the novelty of the presented method more clearly (ll. 52). Methods for emission optimization in the context of inverse modeling of parameters and chemical data assimilation should be noted and shortly discussed with respect to advantages and disadvantages of the new method.

[Response]: We thank the reviewer for the good suggestion about detailing the existing methods and clarifying the novelty of the presented method. In general, the traditional top-down inversion methods use four-dimensional data assimilation (Mendoza-Dominguez and Russell, 2000) or Kalman Filter methods combined with chemical transport model sensitivity analysis, like decoupled direct method in three dimensions (DDM-3D, Napelenok et al., 2008), or adjoint method (Cao et al., 2018), to optimize the gap between the simulation and observation through adjusting the emission from a priori estimate. Different from previous sensitivity based optimization, this study adopted emission-concentration response functions which provide real-time estimates of the concentrations under various emission scenarios. Therefore it can make the adjustment of emissions match with the observation more straightforwardly, avoiding the calculation of the sensitivities. The advantage of the new method is for its ability in well representing the nonlinearity of PM2.5 and O3 to their precursor emissions,

and can assimilate both pollutants simultaneously by keeping the natural linkage (i.e., both pollutants have contributions from common precursors (NOx and VOC), similar atmospheric diffusion/advection transport, and chemical oxidation reactions). To address the "ill-posedness" inversion problem, in this study we used all the observations for multiple pollutants, and also constrained the adjustment of emissions at provincial scale rather than at each single grid cell, which means that we only change of total emissions of each province and keep the same spatial and temporal variation as that in the priori emission. Such design makes the new method exhibit small sensitivity to the variation of observation site number due to the use of prior knowledge of the spatial distribution of emissions, particularly for certain period when observations is not always available across the whole target area; however, the ability to assimilate concentrations at the edge of the control region is limited. Uncertainties associated with the spatial and temporal variations cannot be reduced, which is the disadvantage of the new method (Xing et al., submitted). Nevertheless, the study mainly focuses on the relative change of total emissions over a relatively large region due to the COVID-19, rather than improving the baseline emissions, our new method is more suitable to address such specific needs.

Following the reviewer's suggestion, we have provided more description about the existing top-down method and clarified the novelty of the proposed method in the revised manuscript as follows.

(Line 50) "In general, the traditional top-down inversion methods use four-dimensional data assimilation (Mendoza-Dominguez and Russell, 2000) or Kalman Filter methods (Hartley and Prinn, 1993) combined with sensitivity analysis of chemical transport modeling, like decoupled direct method in three dimensions (Napelenok et al., 2008), or adjoint method (Cao et al., 2018), to optimize the gap between the simulation and observation through adjusting the emission from a priori estimate."

(Line 90) "Different from previous top-down methods that applying sensitivity based optimization, this study adopted emission-concentration response functions which

provide real-time estimates of the concentrations under various emission scenarios. Therefore it can make the adjustment of emissions match with the observation more straightforwardly by avoiding the calculation of the sensitivities. Meanwhile, the natural linkage exists in air pollutants like PM2.5 and O3 since both pollutants have contributions from common precursors (NOx and VOC), similar atmospheric diffusion/advection transport, and chemical oxidation reactions. The advantage of the new method is for its ability in representing the nonlinearity of PM2.5 and O3 to their precursor emissions, thus can assimilate both pollutants simultaneously by keeping the natural linkage. In addition, to address the "ill-posedness" inversion problem, we took advantage of all available observations for multiple pollutants, and constrained the adjustment of emissions at provincial scale rather than at each single grid cell. That means we only change of total emissions of each province but keep spatial and temporal variation the same as that in the priori emissions. Such design makes the new method has small sensitivity to the change of observation sites due to the use of prior knowledge of the spatial distribution of emissions, which is particularly useful for certain period when observations are not always available across the whole region. However, the new method has limited ability to assimilate concentrations at the edge of the control region, and may suffer uncertainties in the spatial and temporal variations which are unable to be adjusted by this method (Xing et al., submitted). Nevertheless, since the study mainly focuses on the relative change of total emissions over the NCP region due to the COVID-19 rather than improving the baseline emissions, thus our new method is more suitable to address such specific purpose."

Reference:

Mendoza-Dominguez, A., & Russell, A. G. (2000). Iterative inverse modeling and direct sensitivity analysis of a photochemical air quality model. Environmental science & technology, 34(23), 4974-4981.

Hartley, D., & Prinn, R. (1993). Feasibility of determining surface emissions of trace gases using an inverse method in a three‐dimensional chemical transport model.

Journal of Geophysical Research: Atmospheres, 98(D3), 5183-5197.

Napelenok, S. L., Pinder, R. W., Gilliland, A. B., & Martin, R. V. (2008). A method for evaluating spatially-resolved NO x emissions using Kalman filter inversion, direct sensitivities, and space-based NO 2 observations.

Cao, H., Fu, T. M., Zhang, L., Henze, D. K., Miller, C. C., Lerot, C., ... & Hendrick, F. (2018). Adjoint inversion of Chinese non-methane volatile organic compound emissions using space-based observations of formaldehyde and glyoxal. Atmospheric Chemistry & Physics, 18(20).

Xing et al., Data assimilation of ambient concentrations of multiple air pollutants using an emission-concentration response modeling framework, under review

[Comment]: 2. What remains somehow unclear is the way how emissions are changed by the response model. Are emissions only corrected locally (i.e. the emissions at the location of the observations) or does the response model consider inverse non-local transport and transformation processes (i.e. correction of emissions at remote locations)? What is the temporal extension of the corrections? Maybe, this can be described more clearly in the manuscript. In case of non-local corrections, it would be interesting to show the spatial patterns of corrections induced by the inversion.

[Response]: The emissions are corrected at the provincial averaged level, not at the location of the observations. The RSM model is designed to link the emissions aggregated by regions (province in this study) with the concentrations at each grid cell. Therefore, not only the local emissions (at observation location) but also the emissions at the surrounded area (the whole studied region) are adjusted to match with the observations. The spatial pattern of the corrections induced by the adjustment is shown as Figure R1. Apparently, the simulated concentrations over the whole studied region were adjusted, in addition to the observation locations.

Since the RSM was originally built based on the 3-D chemical transport model through

multiple-emission scenarios by changing total emissions of controlled regions, both local source and non-local transport and transformation have been considered in the RSM. We corrected the emissions at the stage level (i.e., the period average). A unified change ratio was applied to each pollutant emission for each stage, and the temporal variation such as hourly profiles was kept the same as that in the priori estimate.

We have clarified this point in the revised manuscript as follows.

(Line 115) "We then adjust the total emission ratio of five pollutants (i.e., NO2, VOC, SO2, NH3 and primary PM2.5) in five provinces of NCP (i.e., Beijing, Tianjin, Hebei, Shandong, and Henan) to estimate the updated simulated concentrations to match with the observations. Since the RSM was originally built based on the 3-D chemical transport model through multiple-emission scenarios by changing total emissions at controlled regions, both local source and non-local transport and transformation have been considered in the assimilation."

(Line 181) "The stage-averaged emissions are corrected by applying a unified change ratio to each pollutant emission at each stage, and the temporal variations such as hourly profiles are kept the same as those in the priori estimates."

[Comment]: 3. Concerning the evaluation with observations: In data assimilation, forecasts are usually evaluated by independent observations, which are not considered in the optimization procedure. This is the standard way to investigate the usefulness of applied corrections. The corrected forecasts should fit the observations used for correction for any consistent method by definition. Thus, an evaluation with those observations does not provide additional information in the methodical point of view.

[Response]: This study aims to quantify the change of emissions based on observations, thus all available observations were used for that purpose. However, we agree with the reviewer that the evaluation of the performance of assimilation should be evaluated by using independent set of observation which is different from what was used for assimilation. To address the reviewer's concern, we conduct the cross validation to

examine the performance, by using half of the observation sites randomly selected in each province for correction and the rest half for testing. Since the RSM-based method can help adjust the total emissions following the same spatial within each region, thus it has small sensitivity to the change of observation number, as suggested by the result that the performance of using 50% sites is quite similar to that for using all sites (Figure R2-3).

We have also compared the estimated emission change ratios by using half of the observations. The results suggest that the ratios estimated from 50% sites are also quite close to those estimated from using all sites, as shown in Figure R4.

To address the reviewer's concern, we added more discussion about cross validation in the revised manuscript as follows.

(Line 295) "To evaluate the performance of assimilation, we also conducted the cross valdiation by using 50% observation sites for estimating the emission ratio which to be applied on the rest 50% observation sites for testing. The performance of cross valdiation is exmained, suggesting quite similar results with that using all observation sites as shown in Figure 4. The estimated percent changes in emissions due to the shutdown in Period 2 from cross-validation are also close to that using all observation sites, as shown in Figure S13."

[Comment]: In the description of the correction of SO2-emissions: Are primary SO4 (CspôĂĂĂSO4,ll. 118, Eq. (E4) ) concentrations assumed to be correct? This might be worth mentioning in the description.

[Response]: Yes, here we assume the primary SO4 concentrations to be correct, though it might also suffer uncertainties. We have clarified this point in the revised manuscript as follows.

(Line 154) "Also the primary SO4 concentrations were assumed to be correct."

[Comment]: Maybe related to point 2 (above): How does are the emissions from the

bottomup inventory of 2017 corrected by the response model? Is this also based on the observations used for correction later on? Or are the total annual emissions scaled by some correction values (i.e. constant correction of emissions keeping the annual and diurnal variations constant)?

[Response]: The emissions from the bottom-up inventory of 2017 was used as the prior estimates for both 2019 and 2020 and they were adjusted based on observations using the RSM method. As explained in previous comment, we keep the spatial distribution the same within each province as the bottom-up inventory. The diurnal variations within each day is also the same as those in bottom-up inventory. We only scale the total emissions of each province during each stage based on the observations.

We have clarified this point in the revised manuscript as follows.

(line 169) "Since our study focuses on periods in 2019 and 2020, we first use the response model to adjust the 2017 emission inventory to match the observations during two study periods."

(line 181) "The stage-averaged emissions are corrected by applying a unified change ratio to each pollutant emission at each stage, and the temporal variations such as hourly profiles are kept the same as those in the priori estimates."

[Comment]: It might be worth explaining the estimation of the hypothetical no-shutdown emissions a bit more in detail (ll. 186). Are these estimated from temporally- and spatially averaged ratios between the periods?

[Response]: The hypothetical no-shutdown emissions are estimated based on the temporal profile that no considering the shutdown effects, following the seasonality of each emission sector. It is roughly close to the temporally averaged ratios between the periods, while the exact values depend on the number of days covering in each period. We didn't adjust the spatial distribution of emissions within each province, thus it keeps the same as that in the bottom-up inventory.

We have clarified this point in the revised manuscript, as follows.

(line 217) "The hypothetical no-shutdown emissions for Period 2 (noted as Period 2H) are estimated using ratios of emissions for Period 2 and Period 1 and 3 based on the temporal profile (i.e., reflect the monthly variation across a year) of the bottom-up inventory which only reflects the natural evolution of emissions across a year for each sector. It is roughly close to the temporally averaged ratios between the Period 1 and 3, and the exact values depend on the number of days covering in each period."

[Comment]: As far as I understand, Figure 4 shows averaged observations over the entire plain. Such spatially-averages should be used with caution as observations from single stations might be missing for some time (as noted in ll. 172). This hampers the interpretation of the temporal evolution of the plotted observations. Moreover, it needs to be made clear if the simulated concentrations shown in Fig.4 only refer to these stations, which provide used observations at each time. Drawing a continuous line might be misleading in case the simulated concentrations include different stations at different times.

[Response]: Figure 4 presents the averaged observations over the entire region and the simulated concentrations only refers to the observations at each time. We agree with the reviewer that the observations from single stations might be missing at some point, although we have carefully examined the observation and only used sites that have more than 90% availability during the study period. The continuous line might be misleading.

As the reviewer suggested, we have updated the continuous lines into scattered points to avoid confusing, and clarified this point in the revised manuscript as follows.

(line 202) "Only data at monitoring sites that covered the 90% of entire period is considered."

(Figure 4) "the regional average concentrations were calculated using spatially and

temporally matched simulated and observed values"

[Comment]: The reference to Figure 2 in line 162 is not clear. It seems to be not connected to Fig. 2 of this manuscript. What is the content of the cited manuscript by "Xing et al, under review"?.

[Response]: Sorry for the typo. The Figure 2 represents that figure in the reference. We have clarified this in the revised manuscript as follows.

(line 190) "Specifically, deep-learning technology was used to fit response surfaces for the three months in 2019 and 2020 using CMAQ simulations for baseline and zero-out emissions conditions (see Figure 2 in Xing et al. (2020))."

Reference:

Xing, J., Zheng, S., Ding, D., Kelly, J. T., Wang, S., Li, S., ... & Zhu, Y. (2020). Deep learning for prediction of the air quality response to emission changes. Environmental science & technology, 54(14), 8589-8600.

[Comment]: The x-axis of Figure 4 is not defined. Does this refer to the day of year?

[Response]: Yes, it refers to "days after Jan 1st". We have added the name of x-axis in the revised manuscript.

[Comment]: Fig. 4: The red line is hardly visible e.g. in Figure 4c. It might be useful to plot it as somehow thicker/ dashed line.

[Response]: The red line was overlapped with the green line. As the reviewer suggested, we have replotted the figure by using markers in the revised manuscript.

[Comment]: Fig. 6: If I did not miss it, it would be interesting to include the concentrations resulting from changes in all emissions due to the shutdown (Period 2, incl. Shutdown-effects) in Figure 6 (maybe instead of plotting observations). This would make the overall changes in concentrations due to the shutdown more clear than comparing the simulated no-shutdown concentrations with observations. If this comparison

was made for a specific purpose, maybe did not became clear to me in the text.

[Response]: The Figure 6 presents the individual impact of emission changes that have been scaled based on the ratio of observation to the adjusted simulation after considering overall impacts. Therefore, the overall changes in concentrations due to the shutdown can be reflected by the difference between the observation (OBS) and simulation with no consideration of shutdown (oSIM).

We have clarified this point in the revised manuscript as follows.

(line 346) "One thing should be noted that we scaled the individual impact of emission changes based on the ratio of observation to the adjusted simulation after considering overall impacts, to eliminate the small discrepancy between the observations and the adjusted simulations after considering the overall impacts. Therefore, the overall changes in concentrations due to the shutdown can be reflected by the difference between the observation (OBS) and simulation with no consideration of shutdown (oSIM)."

[Comment]: Line 266: The response to O3 to NOx and VOC appears to be quite linear in the local regime as shown in Fig. 5. I would suggest to replace the formulation in ll. 266 by e.g. "opposite response" or "compensating effects".

[Response]: We agree with the reviewer that at the local area (within small change) the response is quite linear. As the reviewer suggested, we have replaced the word "nonlinear" to be "opposite response" in the revised manuscript.

[Comment]: Line 285: To which decrease in NO3 concentrations is here refereed to? Is this the decrease in Period 2 compared to Period 1?

[Response]: The decrease in NO3 concentration refers to the simulation with no-shutdown against with that with shutdown in Period 2. We have clarified this point in the revised manuscript as follows.

(line 321) "A larger decrease in simulated (from that with no consideration of shutdown influences) than observed NO3- concentrations is associated with the NOx emission

reductions, but the change of NH3 emissions can hardly increase the NO3- concentrations under such strong NH3-rich conditions"

[Comment]: Finally, three small technical suggestions (i) delete the "a" in line 280: "... result in strong NH3-rich conditions, ...", (ii) add an "a" in line 302: "..., we conducted a sensitivity analysis ..." and (iii) in line 340: e.g. "... was applied to the investigation of emission changes ..."

[Response]: As the reviewer suggested, we have fixed these three typos in the revised manuscript.

2019          2020

O₃

sim-org: O3-2019     sim-adj: O3-2019          sim-org: O3-2020     sim-adj: O3-2020

PM₂.₅

sim-org: PM25_TOT-2019   sim-adj: PM25_TOT-2019      sim-org: PM25_TOT-2020   sim-adj: PM25_TOT-2020

**Fig. 1.** Assimilated concentrations of O3 and PM2.5 during Period-2

[Figure]

**Fig. 2.** Cross validation of the assimilation performance (cross validation #1)

[Figure]

**Fig. 3.** Cross validation of the assimilation performance (cross validation #2)

[Figure]

[Figure]

**Fig. 4.** Comparison of estimated percent changes in emissions due to the shutdown in Period 2 from cross-validation (cv1-cross validation #1 by using randomly selected half of the observation sites in each pro

---

## Author Comment (AC2) · 3 Oct 2020

[Comment]: Xing et al. used the response surface model to estimate the emission changes based on the air pollutants concentration changes during COVID-19 in China. Accurate and timely estimate of emission changes are critical to investigate how the air pollutants response to rapid environment changes, such as halt of transportation, slowdown of industry and energy sector during COVID-19, which are missing in recently published journal articles studying the air quality response to COVID-19. The methodology proposed in this study provides a promising framework connect real-time emission changes with abrupt environment changes. I am also very satisfied when the

authors provide hypothetical individual emission changes on the influence of ambient concentration changes (section 3.3), which is very helpful to design the multi-pollutants control strategies in China. The manuscript fits for the journal as well, and I suggest acceptance for this journal.

[Response]: We thank the reviewer for recognition of the implications of the results of the analysis presented, and positive comments.

[Comment]: L162: Fig 2 is not related to the reference pointed here; Also by looking at Fig. 2, there are more observations sites besides NCP. So I suggest the author rewrite the legend for Fig. 2.

[Response]: Sorry for the typo. The Figure 2 represents that in the reference. We have clarified this in the revised manuscript as follows.

(line 192) "Specifically, deep-learning technology was used to fit response surfaces for the three months in 2019 and 2020 using CMAQ simulations for baseline and zero-out emissions conditions (see Figure 2 in Xing et al. (2020))."

We have also rewritten the captain of the Figure 2 as follows.

"Simulation domain and location of observation sites (colorred area: five provinces of North China Plain; red dots: surface monitor sites for NO2, SO2, O3 and PM2.5; blue dots: monitor sites for PM2.5 chemical compoments)"

[Comment]: Fig 3. Consider to put subscript letter for those air pollutants.

[Response]: As the reviewer suggested, we have put subscript letter for those pollutants in the revised manuscript.

[Comment]: Fig 4. Consider to put the simulations with the prior emission (without using the RSM to adjust) for comparisons purpose.

[Response]: As the reviewer suggested, we have put the simulations with the prior emission (without using the RSM to adjust) in the revised manuscript.